# Autocatalytic microtubule nucleation determines the size and mass of *Xenopus laevis* egg extract spindles

Franziska Decker[1,2,3], David Oriola[1,2,3], Benjamin Dalton[1,2,3], Jan Brugués[1,2,3]*

[1]Max Planck Institute of Molecular Cell Biology and Genetics, Dresden, Germany; [2]Center for Systems Biology Dresden, Dresden, Germany; [3]Max Planck Institute for the Physics of Complex Systems, Dresden, Germany

**Abstract** Regulation of size and growth is a fundamental problem in biology. A prominent example is the formation of the mitotic spindle, where protein concentration gradients around chromosomes are thought to regulate spindle growth by controlling microtubule nucleation. Previous evidence suggests that microtubules nucleate throughout the spindle structure. However, the mechanisms underlying microtubule nucleation and its spatial regulation are still unclear. Here, we developed an assay based on laser ablation to directly probe microtubule nucleation events in *Xenopus laevis* egg extracts. Combining this method with theory and quantitative microscopy, we show that the size of a spindle is controlled by autocatalytic growth of microtubules, driven by microtubule-stimulated microtubule nucleation. The autocatalytic activity of this nucleation system is spatially regulated by the limiting amounts of active microtubule nucleators, which decrease with distance from the chromosomes. This mechanism provides an upper limit to spindle size even when resources are not limiting.

DOI: https://doi.org/10.7554/eLife.31149.001

*For correspondence: brugues@mpi-cbg.de

Competing interests: The authors declare that no competing interests exist.

## Introduction

A general class of problems in biology is related to the emergence of size and shape in cells and tissues. Reaction diffusion mechanisms have been broadly successful in explaining spatial patterns in developmental biology as well as some instances of intracellular structures (*Turing, 1952*; *Howard et al., 2011*). The mitotic spindle, a macromolecular machine responsible for segregating chromosomes during cell division, is thought to be a classic example of such reaction diffusion processes. A diffusible gradient of the small GTPase Ran emanating from chromosomes has been shown to trigger a cascade of events that result in the nucleation of microtubules, the main building blocks of the spindle (*Kaláb et al., 2006*; *Caudron et al., 2005*). The spatial distribution of microtubule nucleation is key for understanding size and architecture of large spindles. This is because microtubules in these spindles are short and turnover rapidly in comparison to the entire structure (*Redemann et al., 2017*; *Brugués et al., 2012*; *Needleman et al., 2010*). The mechanisms underlying the spatial regulation of microtubule nucleation, however, are still unclear (*Prosser and Pelletier, 2017*; *Petry, 2016*). One possibility is that the interplay between Ran-mediated nucleation and microtubule turnover governs spindle assembly (*Kaláb et al., 2006*; *Caudron et al., 2005*). However, the role of the Ran gradient in determining spindle size is still controversial. For instance, in cell culture systems, the length scale of the Ran gradient does not correlate with spindle size (*Oh et al., 2016*). A second possibility is that autocatalytic growth accounts for spindle assembly via microtubule-stimulated microtubule nucleation (*Petry et al., 2013*; *Goshima et al., 2008*; *Loughlin et al., 2010*; *Ishihara et al., 2016*). However, autocatalytic mechanisms suffer from the fact that their growth is hard to control. Although autocatalytic growth can be regulated by limiting the catalyst,

**eLife digest** When cells divide, they first need to create a copy of their genetic material, which they then evenly distribute between their daughter cells. This is done by a complex of proteins known as the mitotic spindle, which divides the chromosomes that carry the genetic material in the form of genes. The mitotic spindle is mainly made of tubulin proteins that are arranged to form hollow cable-like filaments, called the microtubules. Microtubules are dynamic structures that can grow or shrink by adding or removing tubulin proteins. Unlike the spindle, which can 'live' up to hours, the microtubules only live for about 20 seconds and need to be constantly renewed to maintain the structure.

To successfully distribute the genetic material, spindles need to have the right length. Previous research has shown that the length of a spindle adapts to the size of a cell – the larger the cells, the larger the spindles. However, in very large cells, such as the cells of an embryo when they first divide, spindles have an upper size limit. It is thought that specific proteins produced by the chromosomes help to regulate the formation of new microtubules and thereby also influence the size of the spindle. However, until now it was not clear how exactly they do so and if this also sets the upper size limit.

To further investigate microtubule renewal and its relation to spindle size, Decker et al. used spindles assembled in cell extracts from the eggs of the African clawed frog. The results showed that the new microtubules grow off the existing ones, like new branches of a tree. The branching happens when the established microtubules interact with specific molecules emitted by the chromosomes, and the concentration of these molecules decreases with distance from the chromosomes. This concentration gradient regulates how many microtubules grow at different distances from the chromosomes and so sets the size of spindles.

These findings help us to understand how biological structures are built out of dynamic and short-lived components. Moreover, a better understanding of how mitotic spindles grow might eventually help to develop new treatments for cancer and other diseases.

DOI: https://doi.org/10.7554/eLife.31149.002

such mechanisms are unlikely to function in the large cells of developing eggs such as *Xenopus*, where resources are not limiting (*Crowder et al., 2015*). Understanding the role of microtubule nucleation in setting the size of spindles is limited by the fact that little is known about the rate, distribution, and regulation of microtubule nucleation in spindles (*Prosser and Pelletier, 2017*; *Petry, 2016*). This is partly because of the lack of methods to measure microtubule nucleation in spindles. Here, we measured microtubule nucleation in spindles assembled in *Xenopus laevis* egg extract using laser ablation. We show that microtubule nucleation is spatially dependent and requires physical proximity to pre-existing microtubules. Our findings are consistent with a theoretical model in which autocatalytic microtubule nucleation is regulated by the amount of the active form of spindle assembly factors. This mechanism provides a finite size for spindles even when resources are not limiting.

## Results

### Microtubule nucleation is spatially regulated

Microtubules grow from the plus ends while minus ends remain stable (*Howard, 2001*). Thus, the location of minus ends functions as a marker for microtubule nucleation. However, in spindles microtubules constantly flux towards the poles (*Mitchison, 1989*), and measuring the location of a microtubule minus end at a particular time does not correspond to its original site of nucleation (*Brugués et al., 2012*). To decouple microtubule transport from microtubule nucleation, we inhibited kinesin-5 (Eg5) in spindles assembled in *Xenopus laevis* egg extracts. This inhibition stops microtubule transport and leads to the formation of radially symmetric monopolar spindles (monopoles) that have a similar size as regular spindles (*Miyamoto et al., 2004*; *Skoufias et al., 2006*) (*Figure 1A*, *Figure 1—figure supplement 1* and *Video 1*). The location of minus ends in these monopoles corresponds to the location of microtubule nucleation.

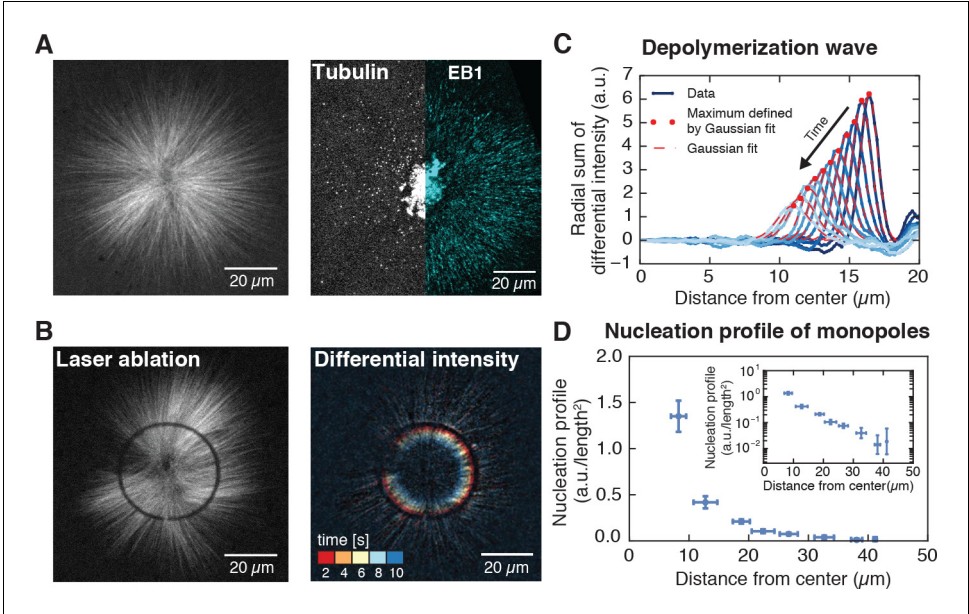

**Figure 1.** Microtubule nucleation in monopolar spindles is spatially regulated. (A) Fluorescence image of a monopolar spindle (left), and single-molecule fluorescent tubulin and EB1-GFP (right). (B) Circular laser cut and corresponding differential intensity depolymerization front at different times. (C) Radial sum of differential intensities at different time points (from dark to light blue) of one cut at a radius of 19 μm from the center. The area under each curve equals the mass of microtubules depolymerized per time interval of 2 s. (D) Nucleation profile of monopolar spindles (N = 117 cuts, mean ± SD).
DOI: https://doi.org/10.7554/eLife.31149.003

The following figure supplements are available for figure 1:

**Figure supplement 1.** Monopolar spindles have a similar size as regular spindles.
DOI: https://doi.org/10.7554/eLife.31149.004

**Figure supplement 2.** The depolymerization velocity of microtubules in monopolar spindles is indistinguishable from the one in spindles.
DOI: https://doi.org/10.7554/eLife.31149.005

**Figure supplement 3.** Depolymerization wave of microtubules cut by laser ablation.
DOI: https://doi.org/10.7554/eLife.31149.006

---

Three independent measurements show that inhibiting microtubule transport does not affect dynamic parameters of microtubules. First, microtubules in these structures polymerize at 20.9 ± 5.1 μm/min (N = 7 monopoles, *Figure 1A* and *Video 1*), which is indistinguishable from the polymerization velocity in spindles, 22.7 ± 8.4 μm/min (N = 4 spindles). Second, microtubules from monopolar and control spindles depolymerize at the same velocity (33.5 ± 6.4 μm/min and 35.9 ± 7.3 μm/min respectively, see *Figure 1—figure supplement 2*). Third, microtubule lifetime distributions of monopolar spindles, measured by single-molecule microscopy of tubulin dimers, give an average lifetime of 19.8 ± 2.2 s, consistent with similar measurements in regular spindles (*Needleman et al., 2010*) (Methods and materials and *Video 2*).

To localize microtubule nucleation events, we measured the density of minus ends in monopolar spindles by analyzing synchronous waves of microtubule depolymerization from laser cuts similar to Ref. (*Brugués et al., 2012*). Briefly, cut microtubules rapidly depolymerize from the newly generated plus ends, while the new minus ends remain stable. The minus end density at the location of the cut can then be obtained from the decrease of the microtubule depolymerization wave, but as opposed to Ref. (*Brugués et al., 2012*), our method resolves the minus end locations with a single laser cut (see *Figure 1B–C*, *Figure 1—figure supplement 3*, *Figure 2—figure supplement 1*, *Video 3*, *Video 4*; a detailed explanation of the method can be found in the Methods and materials and Appendix 1). We define the microtubule nucleation profile at a distance $r$ from the center of the monopole as the number of minus ends per unit length at $r$ divided by $2\pi r$. We measured the

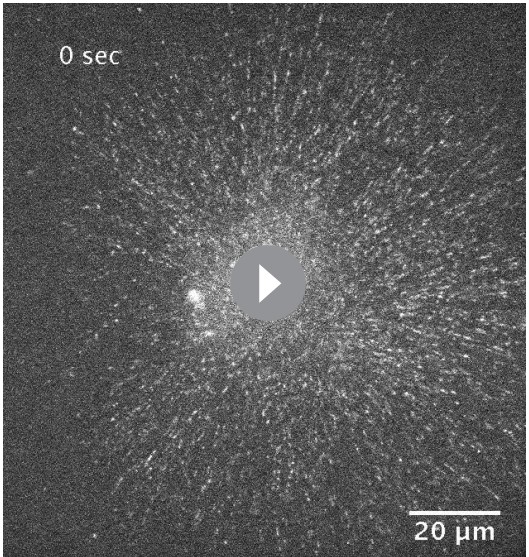

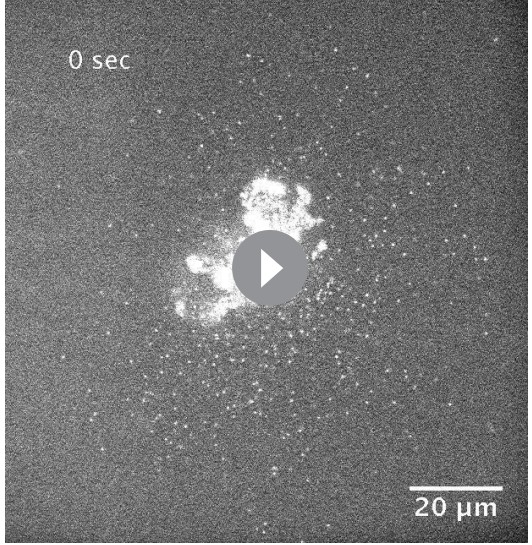

**Video 1.** EB1-GFP comets in monopolar spindles. Pre-assembled monopolar spindles (200 µM STLC) were imaged after adding ~0.2 µg/ml EB1-GFP. Individual frames were recorded every second with subsequent averaging of 3 frames.
DOI: https://doi.org/10.7554/eLife.31149.007

**Video 2.** Tubulin speckles in a monopolar spindle. Pre-assembled monopolar spindles (200 µM STLC) were visualized by adding ~1 nM purified Atto565 frog tubulin. Images were taken every second with subsequent averaging of 4 frames.
DOI: https://doi.org/10.7554/eLife.31149.008

microtubule nucleation profile across the entire structure by performing laser cuts at different distances from the center of the monopoles. These measurements revealed that microtubule nucleation extends throughout monopoles, with the highest nucleation near the center and monotonically decreasing far from the center (see *Figure 1D*), indicating that the strength of microtubule nucleation is spatially regulated.

## Microtubule nucleation depends on the stability of microtubules

Several mechanisms have been proposed to regulate microtubule nucleation. From a biophysical perspective, these mechanisms can be categorized into two scenarios: (i) microtubule-dependent nucleation, in which a pre-existing microtubule stimulates the nucleation of a new microtubule, or (ii) microtubule-independent nucleation, in which factors other than pre-existing microtubules (e.g. diffusible cues in the cytoplasm) stimulate nucleation (*Prosser and Pelletier, 2017*; *Petry, 2016*; *Petry et al., 2013*; *Goshima et al., 2008*; *Clausen and Ribbeck, 2007*; *Ishihara et al., 2014a*; *Carazo-Salas et al., 2001*).

If microtubule nucleation depends on pre-existing microtubules, altering microtubule stability should change the nucleation profile – a microtubule that exists for a longer time would have a higher probability to stimulate the creation of more microtubules. To test this scenario, we increased microtubule stability by inhibiting the depolymerizing kinesin MCAK (*Walczak et al., 1996*) using antibodies. MCAK inhibition led to a dramatic increase in monopole size (see *Figure 2A*). Both the average length and stability of microtubules increased threefold after inhibition (*Figure 2B–C* and *Figure 2—figure supplement 1*) as assessed by laser ablation (8.0 ± 0.3 µm versus 23.6 ± 3.6 µm, see Methods and materials and [*Brugués et al., 2012*]) and single microtubule lifetime imaging (19.8 ± 2.2 s versus 60.4 ± 4.4 s), *Video 2*, *Video 5* and Supplementary *Figure 2*. These measurements are consistent with MCAK modifying the catastrophe rate (*Walczak et al., 1996*; *Tournebize et al., 2000*). We measured microtubule nucleation in this perturbed condition and found that the nucleation profile extends further from the center of the monopole, has a larger amplitude, and decays over a larger distance with respect to control monopoles (*Figure 2D*). Therefore, the number and spatial distribution of nucleated microtubules does indeed scale with microtubule stability in monopolar spindles, which is inconsistent with microtubule-independent nucleation. One possibility is that MCAK-inhibition could by itself increase nucleation independently of microtubules. However, this

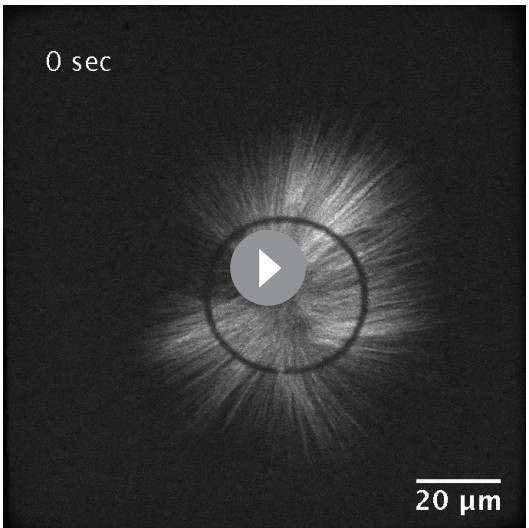 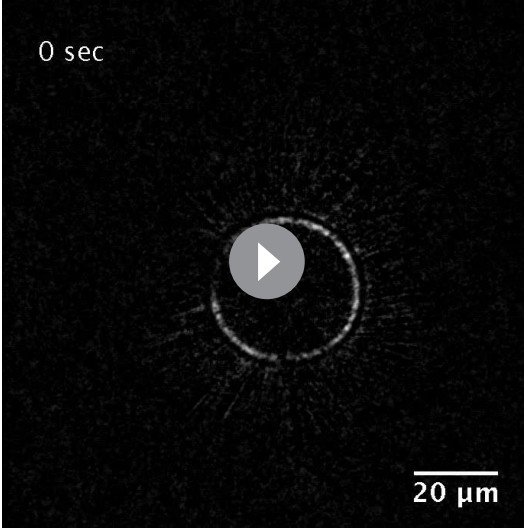

**Video 3.** Depolymerization wave after a cut in a fluorescently labeled monopole. 200 μM STLC, 150 nM Atto565-tubulin. Images were acquired every 500 ms.
DOI: https://doi.org/10.7554/eLife.31149.009

**Video 4.** Fluorescence intensity loss after a cut in a fluorescently labeled monopole as the depolymerization wave propagates. 200 μM STLC, 150 nM Atto565-tubulin. We calculated the differential intensities for a time interval of 2 s. For purposes of visualization, negative intensity values were set to zero.
DOI: https://doi.org/10.7554/eLife.31149.010

would only lead to an overall increase of the amplitude of microtubule nucleation, which alone would not be sufficient to account for the dramatic change in the spatial dependence of the nucleation profile we observe in *Figure 2D*. Thus, microtubule nucleation in these structures depends on the presence and dynamics of microtubules.

## Microtubule nucleation requires physical proximity to pre-existing microtubules

The presence and dynamics of microtubules could alter microtubule nucleation in two ways: microtubules could nucleate indiscriminately in the cytoplasm without requiring microtubules, but their presence concentrates active nucleators through transient interactions with microtubules (*Oh et al., 2016*), or alternatively, microtubules could directly nucleate new microtubules, requiring active nucleators to bind to microtubules to initiate nucleation. In the latter case, the presence of a microtubule is essential for the nucleation process, whereas in the former, microtubules can still nucleate in the absence of microtubules. To test whether microtubule nucleation requires physical proximity to pre-existing microtubules (e.g., a branching process [*Petry et al., 2013*]), we locally blocked microtubule polymerization by adding inert obstacles near the center of monopoles, at locations where nucleation should be expected according to our measurements (*Figures 2D* and *3A*, and *Video 7*). These localized obstacles cannot prevent the diffusion of nucleators, but would prevent microtubules that polymerize towards them to extend further. Consistent with microtubule-stimulated nucleation, the presence of these obstacles inhibited nucleation of new microtubules behind the obstacles, as in a shadow cast by light, whereas microtubules nucleated further around the obstacles, creating a sharp boundary, see *Figure 3A*. These results suggest that monopolar spindles grow to a size larger than an individual microtubule by microtubule-stimulated microtubule nucleation in physical proximity to pre-existing microtubules, which creates an autocatalytic wave of microtubule growth.

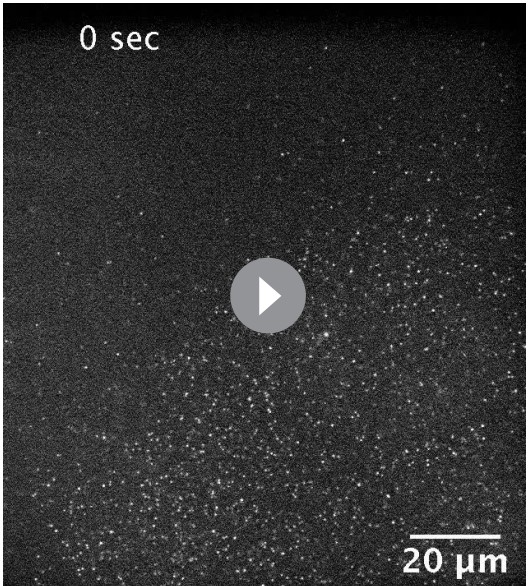

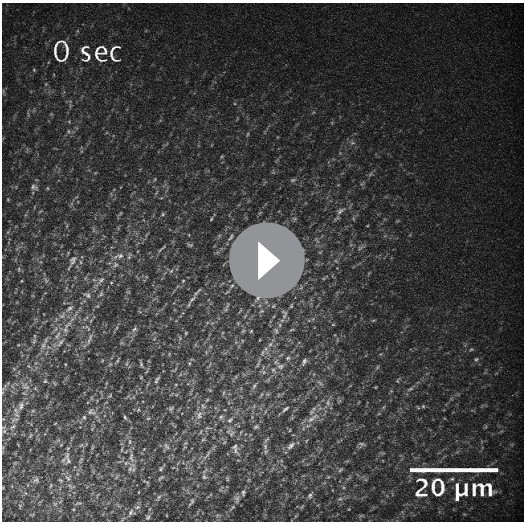

**Video 5.** Tubulin speckles in a monopolar spindle (200 µM STLC) treated with ~30 µg/mg anti-MCAK antibodies. The field of view only shows a quarter of the entire structure. Speckles were created by adding ~1 nM purified Atto565 frog tubulin to pre-assembled structures. In every movie, four frames were averaged during the acquisition.

DOI: https://doi.org/10.7554/eLife.31149.011

**Video 6.** EB1-GFP comets in MCAK-inhibited monopoles. Pre-assembled monopolar spindles (200 µM STLC) were treated with ~30 µg/mg anti-MCAK antibodies and imaged after adding ~0.2 µg/ml EB1-GFP.

DOI: https://doi.org/10.7554/eLife.31149.012

## The amount of active nucleators limits the autocatalytic growth of spindles

For a microtubule structure to have a finite size through an autocatalytic process, each microtubule at the periphery must create on average less than one microtubule at steady state, otherwise the number of microtubules would increase exponentially and the structure would grow unbounded (*Ishihara et al., 2016*). However, measurements of the temporal evolution of microtubule mass in spindles show indeed an initial phase of exponential growth (*Figure 3—figure supplement 1* and (*Clausen and Ribbeck, 2007*; *Dinarina et al., 2009*). This is also consistent with the observation of microtubules creating more than one microtubule on average when inducing bulk microtubule branching by adding TPX2 and constitutively active Ran (RanQ69L) in extracts (*Petry et al., 2013*). These observations raise the question of how spindles reach a finite size through autocatalytic growth (as in the control and MCAK-inhibited monopoles). One possibility is that microtubule dynamics change as a result of limiting amounts of tubulin or microtubule-associated proteins (*Good et al., 2013*; *Hazel et al., 2013*). However, since our cell-free system is not confined, availability of tubulin and microtubule-associated proteins is not limiting. Furthermore, inhibiting MCAK leads to larger monopoles with a microtubule polymerization velocity that is indistinguishable from smaller control monopoles (20.9 ± 5.1 µm/min and 18.8 ± 5.4 µm/min respectively, *Video 6*, *Video 1*, *Figure 3—figure supplement 2* and *Table 1*), suggesting that the availability of tubulin appears not to be diffusion-limited. Finally, microtubule dynamics do not change spatially throughout MCAK-inhibited monopoles (*Figure 3—figure supplement 2*), indicating that spatial variations of tubulin amount or microtubule dynamics cannot explain the finite size of these structures.

Another possibility is that microtubule nucleation is limiting. It has been shown that RanGTP is required for spindle assembly. RanGTP is created only in the vicinity of chromosomes (through the ran nucleotide exchange factor RCC1), which in turn releases spindle assembly factors (SAFs) responsible for nucleating microtubules (*Kaláb et al., 2006*; *Caudron et al., 2005*). Since the active SAFs are naturally limited by their spatially restricted generation, a limiting amount of an active microtubule nucleation factor would therefore be a good candidate as the limiting component for both

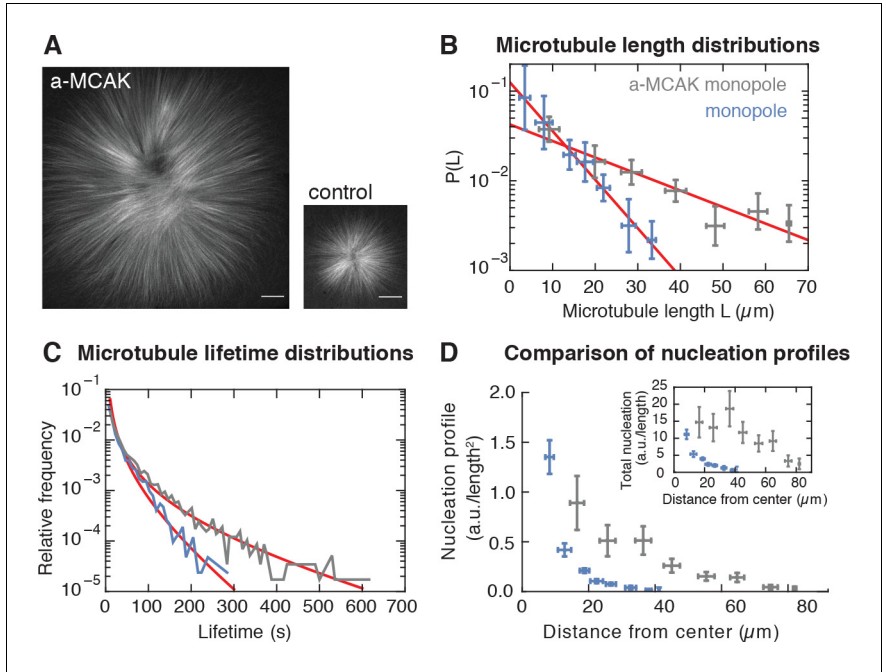

**Figure 2.** Microtubule nucleation depends on the stability of microtubules. (**A**) Inhibition of MCAK leads to larger steady-state monopoles. Scale bar = 20 µm (**B**) Microtubule length distributions measured from laser ablation and fitted to an exponential (mean ± SD). (**C**) Normalized histograms of microtubule lifetimes of control (N = 5331 speckles, five structures) and MCAK-inhibited monopoles (N = 7289 speckles, three structures), and corresponding first-passage time fits (see Methods and materials). (**D**) Nucleation profile of control (N = 117 cuts) and MCAK-inhibited monopoles (N = 74 cuts) in arbitrary units that are consistent in both structures (mean ± SD). The inset shows both nucleation profiles multiplied by the circumference length at each radius, which corresponds to the total microtubule nucleation at that distance from the center of the monopole.
DOI: https://doi.org/10.7554/eLife.31149.013

The following figure supplement is available for figure 2:

**Figure supplement 1.** Microtubule length distributions are independent of the position in monopoles.
DOI: https://doi.org/10.7554/eLife.31149.014

---

autocatalytic growth and size regulation. To test this idea, we added constitutively active Ran (RanQ69L), to pre-existing monopolar spindles. A limiting pool of active nucleators implies that (i) activating nucleators everywhere in the cytoplasm would lead to unbounded microtubule growth in the monopole (similar to large interphase asters in embryos [*Wühr et al., 2010*]), and (ii) new microtubules should nucleate from the pre-existing microtubules of the structure. Adding RanQ69L to pre-existing monopoles immediately started nucleation of new microtubules preferentially at the edge of the pre-existing structures in a wave-like fashion, consistent with microtubule-stimulated growth (*Figure 3B* and *Videos 8* and *9*). This result further suggests that other limiting components that regulate microtubule dynamics alone cannot account for this growth. Taken together, these measurements show that the amount of active nucleators, which is limited by the availability of RanGTP, limits the size of monopolar spindles and is responsible for the bounded growth of these structures.

## Autocatalytic microtubule nucleation model

To test whether a limited pool of active nucleators can quantitatively account for the size and microtubule nucleation in these structures, we developed a biophysical model of autocatalytic microtubule nucleation (see *Figure 4A* and Appendix 1). In our model, inactive nucleators are present throughout the cytoplasm and can be activated at the surface of chromosomes, which is a simplification of the activation of SAFs by RanGTP. The total amount of active nucleators depends on the balance between the rate of activation at the chromosomes and the rate of inactivation (accounting for

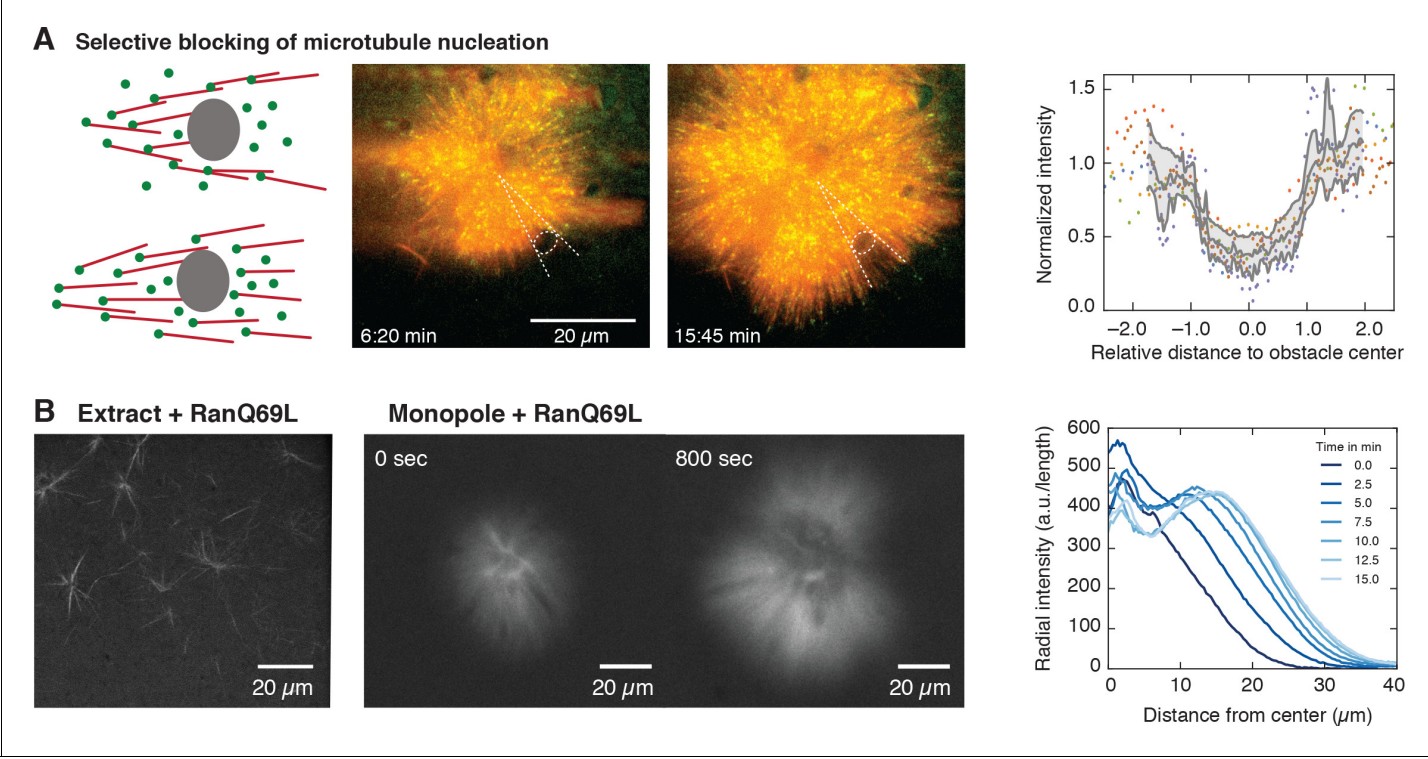

**Figure 3.** Microtubule nucleation requires physical proximity to pre-existing microtubules. (**A**) Inert obstacles (fluorocarbon oil microdroplets or polystyrene beads) immobilized to coverslips selectively block microtubule nucleation. Left: Schematic outcomes depending on whether new microtubules are nucleated in physical proximity from pre-existing microtubules (top) or not (bottom). Middle: Two time points of a microtubule structure with labeled tubulin (red) and EB1-GFP (green) growing around immobilized frog sperm chromosomes. The oil microdroplet is highlighted with a dashed ellipse. Right: Normalized line profiles of shadow regions behind six different obstacles (colored dots). The intensity profiles were normalized to the average intensity outside the obstacles and rescaled to the radius of the obstacle. Gray lines show mean ± SD. (**B**) Left: Addition of RanQ69L to extract homogeneously activates nucleation and creates mini asters after ~20 min (22). Middle: Addition of RanQ69L to monopoles leads to immediate growth of new microtubules from the pre-existing monopoles. Right: Quantification of radial fluorescence intensity profiles at different time points of the growing monopole shown in the middle.

DOI: https://doi.org/10.7554/eLife.31149.015

The following figure supplements are available for figure 3:

**Figure supplement 1.** Growth of a monopolar spindle.
DOI: https://doi.org/10.7554/eLife.31149.016
**Figure supplement 2.** Microtubule dynamics are the same within the entire MCAK-inhibited monopolar spindle.
DOI: https://doi.org/10.7554/eLife.31149.017

sequestration, hydrolysis, or other processes). Once activated at the chromosomes, nucleators can diffuse in the cytoplasm, bind, and unbind from microtubules. When bound to microtubules, active nucleators can nucleate new microtubules at a certain rate, and the newly nucleated microtubules maintain the same polarity as the mother microtubule (*Petry et al., 2013*). This process leads to an autocatalytic wave as a consequence of the self-replicating activity of an extended object. In contrast

**Table 1.** Measurements of microtubule dynamics in spindles and monopoles.

| Microtubule dynamics | Spindle | Monopole | MCAK-inhibited monopole |
|---|---|---|---|
| Lifetime (s) | 16 ± 2* | 20 ± 2 | 60 ± 4 |
| Polymerization velocity (µm/min) | 23 ± 8 | 21 ± 5 | 19 ± 5 |
| Depolymerization velocity (µm/min) | 36 ± 7 | 33 ± 6 | 46 ± 8 |

* Reference **Needleman et al., 2010**.
DOI: https://doi.org/10.7554/eLife.31149.023

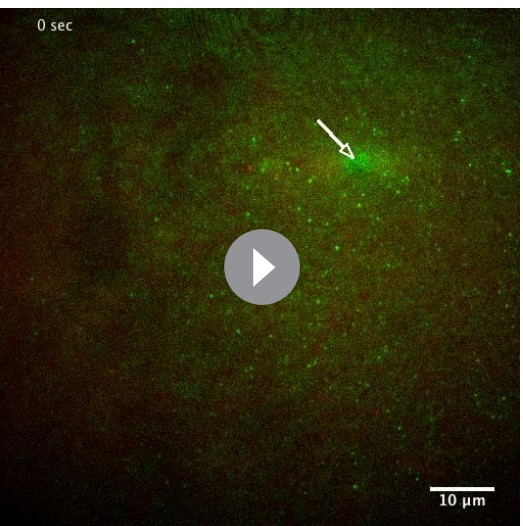

**Video 7.** Monopole growing against an obstacle. The microtubule structure is labeled with tubulin Atto565 (red) and EB1-GFP (green) and grows around immobilized frog sperm chromosomes. The bead is attached to the surface and highlighted with an arrow.
DOI: https://doi.org/10.7554/eLife.31149.018

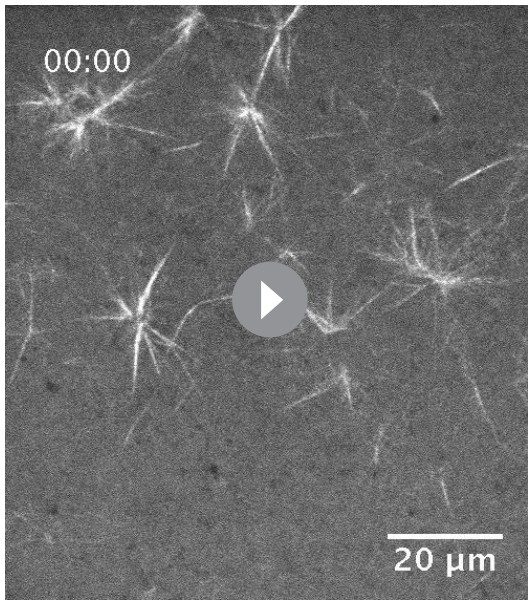

**Video 8.** Ran asters after addition of 30 μM RanQ69L to extract. Microtubules were labeled with 150 nM Atto565-tubulin. Movie starts ∼20 min after adding RanQ69L.
DOI: https://doi.org/10.7554/eLife.31149.019

to a reaction diffusion process, the front propagation is independent of microtubule diffusion and only depends on microtubule dynamics.

In our model, the amount and dynamics of active nucleators are the same for both control and MCAK-inhibited monopoles, leading to the prediction that the two microtubule density profiles would only differ in a parameter controlling the microtubule length or lifetime (see Appendix 1). In particular, both profiles should scale to each other without any fitting parameters by changing the microtubule length as measured independently by laser ablation. To test this prediction, we measured the radial profile of microtubule density of control and MCAK-inhibited monopoles (*Figure 4B*): These microtubule density profiles are qualitatively different –the density of MCAK-inhibited monopoles increases initially and decreases after reaching a maximum, whereas the control monopole decreases monotonically from the origin. Remarkably, both profiles collapse into each other after the parameter-free rescaling of the MCAK-inhibited monopole predicted by the model (see Appendix 1 and *Figure 4C*). To test the model beyond scaling, we fit the MCAK-inhibited profile with two independent parameters and an arbitrary amplitude of the density profile, which agrees quantitatively with the data (see Appendix 1 and *Figure 4B*). By fixing all parameters to the values obtained by this fit (which are the same for the control monopole, see Appendix 1, *Table 2*) and using the measured average microtubule length for the control monopole (Methods and materials, *Table 1*), the model predicts the control monopole microtubule profile. Finally, we can also predict the MCAK-inhibited and control microtubule nucleation profiles from the fitted parameters up to an arbitrary amplitude (common for both profiles) (*Figure 4D*). Remarkably, this prediction is also consistent with flux-corrected microtubule nucleation in regular spindles obtained by laser ablation (see Methods and materials, *Figure 4D* green circles, *Videos 10* and *11*), showing that the same nucleation mechanism holds for regular spindles. Thus, our model for autocatalytic microtubule nucleation accounts for both the microtubule density and nucleation profiles.

## Discussion

Our data and model are consistent with an auto-catalytic mechanism in which microtubule-stimulated microtubule nucleation controls growth in *Xenopus laevis* egg extract spindles. This process is spatially regulated by a gradient of active nucleators that is established by the interplay between the Ran gradient and microtubule dynamics. Microtubules regulate the nucleator activity because they act as the substrate where active nucleators need to bind to nucleate microtubules. Chromatin acts as a trigger for an auto-catalytic wave of microtubule nucleation, and at the same time limits spindle size by controlling the amount of active nucleators through RanGTP. This suggests that the amount of active Ran can tune spindle length, and resolves its controversial relation to spindle length regulation: while a diffusion and inactivation process has a characteristic length scale independent of the amplitude of the gradient – set by the ratio of the squared root of the diffusion and inactivation rate – here we show that both the length scale and amplitude of the gradient of nucleators are involved in regulating the size and mass of spindles. Since

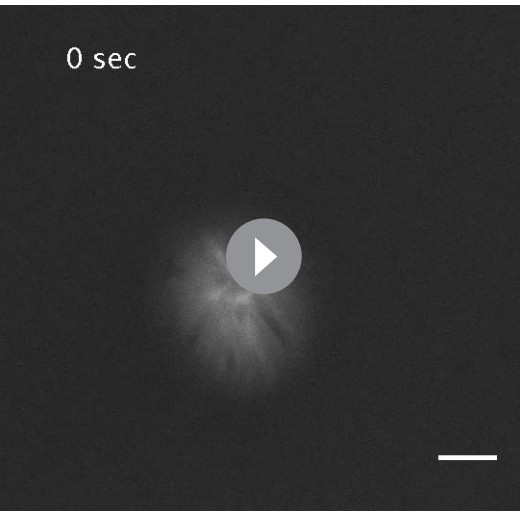

**Video 9.** Growth of pre-existing monopoles after addition of 10 µM RanQ69L. Monopolar spindles (200 µM STLC, labeled with 150 nM Atto565-tubulin) were imaged on a PLL-g-PEG passivated slide immediately after adding RanQ69L. Scale bar = 20 µm.
DOI: https://doi.org/10.7554/eLife.31149.020

the length scale of the gradient is amplified by microtubule-stimulated nucleation, the relevant length scale for setting the size is the distance at which a microtubule generates one or fewer microtubules. Our proposed mechanism therefore allows regulation of spindle size and mass by two means, although microtubule nucleation is the principal control parameter, microtubule dynamics can still fine tune the spindle length (*Reber et al., 2013*). Although our results are restricted to *Xenopus laevis* spindles, we hypothesize that a similar mechanism may also apply to other spindles with a large number of microtubules. This would be consistent with the fact that components involved in microtubule branching have been identified in many eukaryotic systems (*Dasso, 2002*; *Hsia et al., 2014*; *Sánchez-Huertas and Lüders, 2015*). However, further experiments are needed to test this hypothesis.

An autocatalytic nucleation process implies that microtubule structures are capable of richer dynamical behaviors than those arising from the classic view of random nucleation in the cytoplasm via a diffusible gradient. Beyond producing finite-sized structures like spindles and ensuring that new microtubules keep the same polarity as the pre-existing ones, it also allows for a rapid switch into unbounded wave-like growth if nucleators become active throughout the cytoplasm. Indeed, the growth of large interphase asters has been hypothesized as a chemical wave upon Cdk1 activation (*Chang and Ferrell, 2013*; *Ishihara et al., 2014b*). These properties, characteristic of excitable media, provide a unified view for the formation of spindles and large interphase asters in embryos (*Ishihara et al., 2014a*) within a common nucleation mechanism. However, microtubule nucleation differs from regular autocatalytic processes in reaction-diffusion systems such as Fisher-waves and Turing mechanisms (*Turing, 1952*; *Fisher, 1937*) in that its growth does not rely on diffusion or advection. Instead, the process of branching displaces the center of mass of the structure. Thus, it emerges as consequence of the finite extension and dynamics of the reactant (microtubules). The interplay between autocatalytic growth and fluxes driven by motors could lead to general principles of pattern formation and cytoskeletal organization in cells.

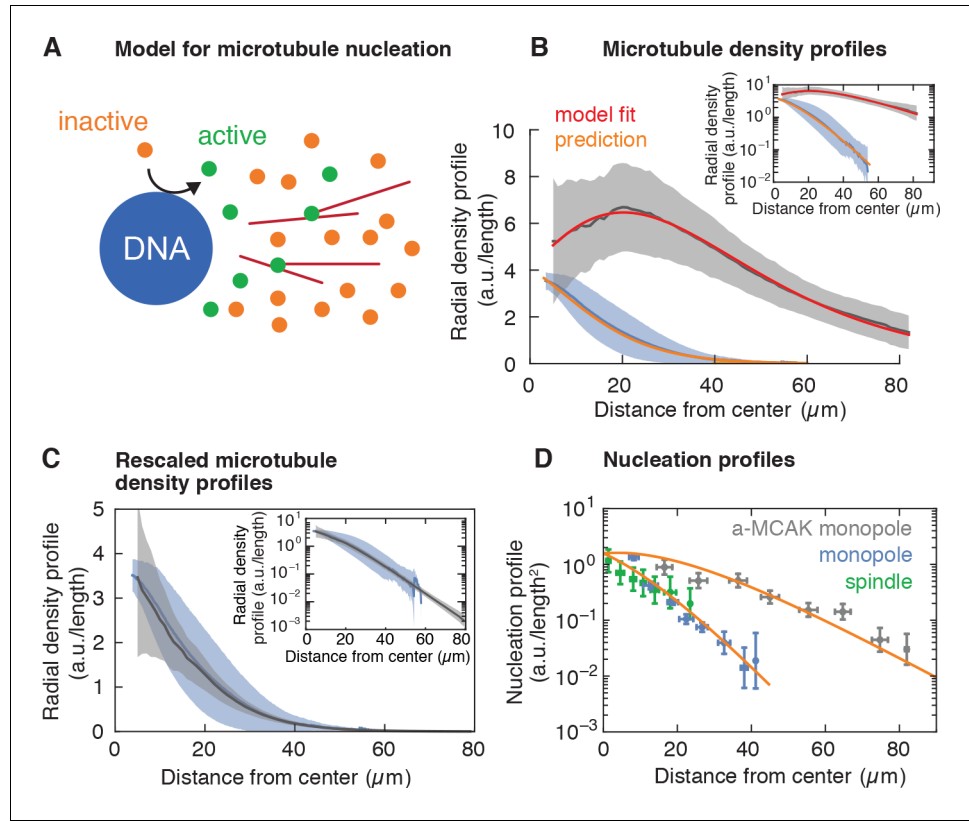

**Figure 4.** Model for microtubule-stimulated nucleation. (**A**) Inactive nucleators (orange circles) are activated around DNA. As active nucleators diffuse (green circles), they can bind and unbind microtubules (red lines). Once bound, they can nucleate a new microtubule with a certain probability. Active nucleators become inactive at a constant rate. (**B**) Radial microtubule density profiles measured from fluorescent images (mean ± SD, $N_{monopoles}$ = 40 (blue), $N_{a-MCAK}$ = 18 (gray)) and corresponding model fit to the MCAK-inhibited and prediction to control monopoles (see Appendix 1). The ratio of the microtubule densities for control and MCAK-inhibited monopoles was determined using structures from the same extract reaction. (**C**) Parameter-free rescaling of the microtubule density profiles predicted by the model: $\rho_C = \rho_M \exp[(1/\ell_M - 1/\ell_C)x]$, where $\rho_C$, $\ell_C$ and $\rho_M$, $\ell_M$ are the density and length of microtubules for the control and MCAK-inhibited structures, respectively, and $x$ is the distance from the center of the structure. In the graph, blue corresponds to the density profile of control monopoles and gray to the rescaled density profile of MCAK-inhibited monopoles. (**D**) Data and predictions (orange) for the nucleation profiles of control (blue) and MCAK-inhibited monopoles (gray) up to a global nucleation amplitude, and flux-corrected regular spindles (green) (mean ± SD, $N_{control}$ = 117 , $N_{a-MCAK}$ = 74, $N_{spindle}$ = 36 cuts).

DOI: https://doi.org/10.7554/eLife.31149.021

The following figure supplement is available for figure 4:

**Figure supplement 1.** Example of the spatio-temporal evolution of autocatalytic microtubule nucleation with spatially varying branching nucleation rate $k_{bra}$, polymerization velocity $v_p$ and turnover rate $\Theta$.

DOI: https://doi.org/10.7554/eLife.31149.022

# Methods and materials

## Cytoplasmic extract preparation, spindle assembly and biochemical perturbations

Cytostatic factor (CSF)-arrested *Xenopus laevis* egg extract was prepared as described previously (*Hannak and Heald, 2006*; *Murray, 1991*). In brief, unfertilized oocytes were dejellied and crushed by centrifugation. After adding protease inhibitors (LPC: leupeptin, pepstatin, chymostatin) and cytochalasin D (CyD) to a final concentration of 10 µg/ml each to fresh extract, we cycled single reactions to interphase by adding frog sperm (to 300–1000 sperm/µl final concentration) and 0.4 mM Ca²⁺ solution, with a

**Table 2.** Parameters used in the model of autocatalytic microtubule nucleation.

| Parameter | Symbol | Value | Procedure |
|---|---|---|---|
| Microtubule turnover rate control monopole ($s^{-1}$) | $\Theta_C$ | $0.05 \pm 0.01$ | Single molecule microscopy |
| Microtubule turnover rate MCAK-inhibited monopole ($s^{-1}$) | $\Theta_M$ | $0.016 \pm 0.001$ | Single molecule microscopy |
| Microtubule length control monopole ($\mu m$) | $\ell_C$ | $8.0 \pm 0.3$ | Laser ablation |
| Microtubule length MCAK-inhibited monopole ($\mu m$) | $\ell_M$ | $24 \pm 4$ | Laser ablation |
| Length scale gradient unbound nucleators ($\mu m$) | $\ell_u$ | $24.9 \pm 0.5$ | Fit to the model |
| Branching parameter (dimensionless) | $\alpha$ | $2.38 \pm 0.01$ | Fit to the model |
| Microtubule density at the center of the monopole (a.u.) | $\rho(0)$ | $4058 \pm 64$ | Fit to the model |

DOI: https://doi.org/10.7554/eLife.31149.024

subsequent incubation of 1.5 hr. While fresh CSF extract containing LPC and CyD was kept on ice, all incubation steps were performed at 18–20°C. The reactions were driven back into metaphase by adding 1.3 volumes of fresh CSF extract (containing LPC and CyD). Spindles formed within 1 hr of incubation. To inhibit kinesin-5 (Eg5) in spindles, S-Trityl-L-Cysteine (STLC) was added to the reactions to a final concentration of 200 µM . Transitions to monopolar spindles were observed within 30–60 min of incubation. To inhibit the depolymerizing kinesin MCAK in monopolar spindles, we added anti-MCAK antibodies to a final concentration of ~30 µg/ml (kind gift from R. Ohi). MCAK-inhibited structures reached their steady-state after ~20 min. Alternatively, we added RanQ69L (kind gift from K. Ishihara) to pre-formed monopoles to a final concentration of 30 or 10 µM and imaged immediately. In the control reactions, the same concentrations were added to extract reactions in the absence of pre-formed structures and imaged after ~20 min incubation. The lower the RanQ69L concentration the later Ran asters formed. Conversely, if a pre-existing structure was present, microtubule nucleation immediately started at the periphery with subsequent growth of the structure. The growth of pre-existing monopolar spindles stopped with the appearance of Ran asters in bulk (after ~20 min depending on the concentration of RanQ69L), consistent with the sequestering of the additional nucleators activated by RanQ69L. Prior to imaging, Atto565 labeled purified porcine tubulin (purified according to Ref. [*Castoldi and Popov, 2003*]) and Höchst 33342 were added to the reactions to a final concentration of 150 nM and ~16 µg/ml, respectively, to visualize microtubules and DNA.

## Image acquisition

Control and MCAK-inhibited monopolar spindles were imaged using a Nikon spinning disk microscope (Ti Eclipse), an EMCCD camera (Andor iXon DU-888 or DU-897), a 60 × 1.2 NA water immersion objective, and the software AndorIQ for image acquisition. The room was kept at constant 20°C. Monopolar spindles after the addition of RanQ69L were imaged using a Nikon wide-field epifluorescence microscope (Ti Eclipse), an sCMOS camera (Hamamatsu Orca Flash 4.0), and a 20 × 0.75 NA objective. In this case, image acquisition was performed using µManager (*Edelstein et al., 2014*). The growth of microtubule structures in the presence of obstacles was imaged using a Nikon total internal reflection fluorescence (TIRF) microscope (Ti Eclipse), equipped with an Andor iXon3 DU-897 BV back-illuminated EMCCD camera, a 100 × 1.49 NA oil immersion objective, and the Nikon software NIS elements.

## Laser cutting procedure and image analysis

The femtosecond laser ablation setup was composed of a mode-locked Ti:Sapphire laser (Coherent Chameleon Vision II) oscillator coupled into the back port of the Nikon spinning disk microscope and delivering 140 fs pulses at a repetition rate of 80 MHz. Cutting was performed using a wavelength of 800 nm and typically a power of 150 mW before the objective. The sample was mounted on a piezo stage that positioned the sample in 3D with sub-micrometer precision. The laser cutting process was automatically executed by a custom-written software that controlled the mechanical shutter in the beam path and moved the piezo stage to create the desired shape of the cut. Lines and circular cuts were performed in several planes to cover a total depth of ~1–2 µm around the focal plane. We adapted the size and geometry of the cut shapes to each spindle or monopolar structure. Cutting was finished within 2 s. Images were acquired at least every 0.5 s during the cutting procedure as well as for ~1 min after the cut. The depolymerization wave typically disappeared within 30 s. Each microtubule structure was cut only once.

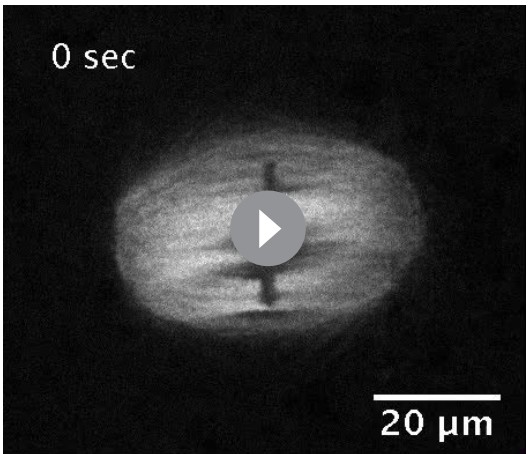
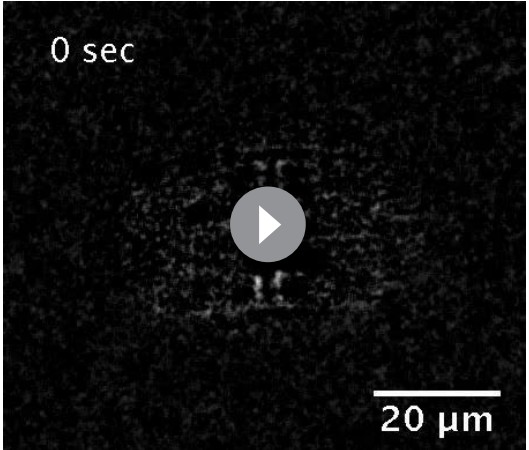

**Video 10.** Depolymerization waves after a cut in a fluorescently labeled spindle. 150 nM Atto565-tubulin. Images were acquired every 500 ms.
DOI: https://doi.org/10.7554/eLife.31149.025

**Video 11.** Fluorescence intensity loss after a cut in a fluorescently labeled spindle as the depolymerization waves propagate. 150 nM Atto565-tubulin. We calculated the differential intensities for a time interval of 2 s. For visualization purposes, negative intensity values were set to zero.
DOI: https://doi.org/10.7554/eLife.31149.026

We analyzed the depolymerization waves using a custom-written Python code. Briefly, for a given cut at position $r$, we subtracted the intensities of images (raw data) with a time difference $\delta t$ of 2–3 s to get the differential intensities $I(x, \phi, t; r)$, where $x$ is the radial coordinate, $\phi$ is the angle, and $t$ is the time after the cut (see *Figure 1B* and *Figure 1—figure supplement 3*). $I$ corresponds to the quantity of microtubules that depolymerized during the time interval $\delta t$. Next, we integrated the differential intensities over $\phi$ and plotted these integrals with respect to the radial coordinate $x$. The depolymerization wave appears as a peak that is traveling towards the center of the monopole and broadening over time (see *Figure 1C*). We fitted Gaussians to these peaks and plotted the area under these Gaussians over time $\tilde{A}(t, r)$ (see Laser ablation method and *Figure 1—figure supplement 3C*). We fitted an exponential to the area decays over distance from the cut, and normalized the decays by the amplitude at the location of the cut. The slope at the position of the cut is proportional to the number of minus ends at this location (see Laser ablation method). To take the local microtubule density into account, we multiplied the normalized slopes at the position of the cut by the averaged angular integral of the microtubule fluorescence intensity at this position. This gives the amount of minus ends per unit length $n_c(y, r)$ at $y$ given a cut performed at $r$. To obtain the two dimensional minus end density (number per unit length squared), we divided by $2\pi r$, which corresponds to the nucleation profile (notice that the nucleation profile has arbitrary units). Averaged microtubule density profiles were obtained from 82 and 12 fluorescence profiles of monopoles and MCAK-inhibited monopoles, respectively. Additionally, we used angular fluorescence profiles of control and MCAK-inhibited structures from the same extract reaction to determine the ratio between these two nucleation profiles and enable a reliable comparison. Finally, in order to obtain the microtubule length distribution, we fitted an exponential function to $n_c(y, r)$ as a function of the cut distance $r$. The slope of $n_c(y, r)$ at $r$ is proportional to the number of microtubules with minus ends at $y$ and plus ends at $r$ (see Appendix 1), which after normalization gives the microtubule length distribution.

Laser cuts in bipolar spindles were similarly analyzed. Instead of circular cuts, we performed linear cuts perpendicular to the long axis of the spindle, which induced two depolymerization waves traveling towards the poles (due to the mixed polarity of microtubules). The waves were analyzed by integrating the differential intensities along the direction of the cut and plotting these integrals as a function of spindle length. This again lead to the depolymerization waves appearing as peaks that are traveling towards

the poles and broadening over time. The subsequent analysis is exactly the same as for monopolar spindles continuing by fitting Gaussians to these peaks as described above.

## Analysis of microtubule dynamics

The microtubule polymerization velocity was measured by adding EB1-GFP to extract reactions to a final concentration of ~0.2 μg/ml and analyzing kymographs drawn along the growth direction of a microtubule (40 kymographs from seven control monopoles obtained from different reactions on two different days, 68 kymographs from five MCAK inhibited monopoles obtained from different reactions on the same day). Microtubule depolymerization velocities were obtained by analyzing the velocity of the fronts after the laser cuts. The maxima of the fitted Gaussians (see Laser cutting procedure and image analysis) were used to determine the position of the depolymerization front as a function of time, which was fitted to a linear function. The slope corresponded to the depolymerization velocity of the cut microtubules, which was found to be constant for each laser cut. We measured microtubule lifetimes by adding Atto565 purified frog tubulin (purified according to Ref. [*Groen and Mitchison, 2016*]) to a final concentration of ~1 nM and subsequent tracking of the speckles using the MOSAIC suite, (*Sbalzarini and Koumoutsakos, 2005*) (5331 speckles from five monopoles from different reactions of 3 different days, 7289 speckles from 3 MCAK-inhibited monopoles from different reactions of the same day). We included only those speckles that appeared and disappeared during the length of the movie (~10 min). To calculate the average lifetime of microtubules, we used the lifetime distribution $\mathcal{P}(t)$ of a diffusion and drift process to fit it to our data according to $\mathcal{P}(t) \sim t^{-3/2}e^{-t/\tau}$, where $\tau/4$ is the expected lifetime of a microtubule of average length, Ref. (*Bicout, 1997*; *Needleman et al., 2010*). A summary of the different measured values is given in *Table 1*.

## Obstacle assay to block microtubule nucleation

Coverslips were cleaned by sonication in 2% Hellmanex and used to assemble parafilm channels of ~3 mm width. Every step of the assay was completed by an incubation at room temperature (10 min up to several hours) and washing of the channel with BRB80 (80 mM PIPES, 1 mM $MgCl_2$, 1 mM EGTA). Channels were subsequently filled with anti-biotin antibodies, Puronic F-127 to block the remaining surface, biotinylated *Xenopus laevis* sperm, biotinylated fluorocarbon oil microdroplets (produced as described in Ref. [*Lucio et al., 2015*]) or biotinylated polystyrene beads acting as inert obstacles, and freshly prepared extract including Atto565 labeled purified porcine tubulin (150 nM final), EB1-GFP (~0.2 μg/ml final), and sodium orthovanadate (0.5 mM final concentration). Image acquisition was performed on a TIRF microscope.

## Measuring the microtubule mass over time

To measure the microtubule mass over time, we added frog sperm to extract and immediately started to acquire z-stacks around the DNA over time. After subtracting the background, we integrated the fluorescence intensity of the labeled microtubules over all z-planes and plotted it as a function of time.

## Passivation of coverslips with PLL-g-PEG

Passivation of coverslips with Poly-L-lysine-g-polyethylene glycol (PLL-g-PEG) was performed according to Ref. (*Field et al., 2017*). In brief, coverslips were placed in a drop of 0.1 mg/ml PLL-g-PEG in 10 M HEPES pH 7.4 on Parafilm for 20 min at room temperature. They were then washed three times in distilled water and dried with a nitrogen jet.

## Acknowledgements

We acknowledge A A Hyman, M Loose, S Grill, T Quail, K Ishihara, R Farhadifar, G Pigino, F Jug, C Norden, F Jülicher and I Patten for fruitful discussions and careful revision of the manuscript. We also thank J Rosenberger for helping on the coding for laser ablation. We kindly thank R Ohi and K Ishihara for providing us the anti-MCAK antibodies and RanQ69L, respectively. We thank Heino Andreas for frog maintenance and the Light Microscopy Facility (LMF) at MPI-CBG for providing TIRF microscopy. This work was supported by the Human Frontiers Science Program (CDA00074/

2014, to JB), EMBO (ALTF 483–2016, to DO), the ELBE postdoctoral program (BD), and a DIGS-BB fellowship provided by the DFG (FD).

## Additional information

### Funding

| Funder | Grant reference number | Author |
|---|---|---|
| Human Frontier Science Program | CDA00074/2014 | Jan Brugués |
| European Molecular Biology Organization | ALTF 483-2016 | David Oriola |
| Deutsche Forschungsgemeinschaft | | Franziska Decker |
| Max-Planck-Gesellschaft | Open-access funding | Franziska Decker David Oriola Benjamin Dalton Jan Brugués |

The funders had no role in study design, data collection and interpretation, or the decision to submit the work for publication.

### Author contributions

Franziska Decker, Conceptualization, Data curation, Software, Formal analysis, Investigation, Visualization, Methodology, Writing—original draft, Writing—review and editing; David Oriola, Formal analysis, Methodology, Writing—review and editing; Benjamin Dalton, Investigation, Methodology; Jan Brugués, Conceptualization, Resources, Data curation, Software, Formal analysis, Supervision, Visualization, Methodology, Writing—original draft, Project administration, Writing—review and editing

### Author ORCIDs

Franziska Decker http://orcid.org/0000-0002-3241-6575
David Oriola http://orcid.org/0000-0002-8356-7832
Jan Brugués http://orcid.org/0000-0002-6731-4130

### Ethics

Animal experimentation: All animals were handled according to the directive 2010/63/EU on the protection of animals used for scientific purposes, and the german animal welfare law under the license document number DD24-5131/367/9 from the Landesdirektion Sachsen (Dresden) - Section 24D.

### Decision letter and Author response

Decision letter https://doi.org/10.7554/eLife.31149.031
Author response https://doi.org/10.7554/eLife.31149.032

## Additional files

### Supplementary files

• Transparent reporting form
DOI: https://doi.org/10.7554/eLife.31149.027

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

## Appendix 1

DOI: https://doi.org/10.7554/eLife.31149.028

### Laser ablation method

We quantify the total amount of microtubule depolymerization as a function of time $t$ from a cut at a distance $r$ from the center of the monopolar spindle between two frames separated by $\delta t$ by summing the total differential intensity (see **Figure 1B,C**, and **Figure 1—figure supplement 1**),

$$\tilde{A}(t,r) = n_d(t,r)v_d\sigma_f\delta t, \tag{1}$$

where $n_d(t,r)$ is the total number of depolymerizing microtubules from the cut at $r$, $v_d$ the depolymerization velocity and $\sigma_f$ the fluorescence per unit length of microtubule. We define the total microtubule depolymerization rate by $A(t,r) = \tilde{A}(t,r)/\delta t$. The integrated differential intensity along the angular coordinate $\phi$ has a very well defined peak along the $x$ radial coordinate that moves as a function of time (see **Figure 1C** and **Figure 1—figure supplement 1**), allowing to express the total microtubule depolymerization rate as a function of the position of the peak $y(t)$ with respect to the monopole center, $A(y,r)$ (see **Figure 1—figure supplement 1**). As the front moves, the total microtubule depolymerization rate decreases (see **Figure 1C**). The derivative with respect to the position of the front is:

$$\frac{dA(y,r)}{dy} = -\frac{dn_d(y,r)}{dy}v_d\sigma_f = n_c(y,r)v_d\sigma_f. \tag{2}$$

Where $n_c(y,r)$ is the number of minus ends per unit length at a distance $y$ from the center, from microtubules cut at a distance $r$. Therefore,

$$n_c(y,r) = \frac{1}{v_d\sigma_f}\frac{dA(y,r)}{dy} \tag{3}$$

The number of minus ends per unit length $n_c(y,r)$ is related to the number density of microtubules $\rho(y,\eta)$, with minus ends at $y$ and plus ends at $\eta$ (number per unit length squared), by:

$$n_c(y,r) = \int_r^\infty \rho(y,\eta)d\eta \tag{4}$$

The actual number of minus ends (number per unit length) at a position $r$, $n(r)$, which is proportional to the total microtubule nucleation at that location in the absence of transport, is by definition $\int_r^\infty \rho(r,\eta)d\eta$, where we integrate for all possible lengths of microtubules with minus ends at $r$. This quantity is related to the number of minus ends of microtubules cut at $r$ evaluated at the position of the cut, $n_c(r,r)$:

$$n(r) \equiv \int_r^\infty \rho(r,\eta)d\eta = n_c(r,r) \tag{5}$$

In order to obtain the two-dimensional density of minus ends (or nucleation profile) in the polar geometry of the monopole, we divide $n(r)$ by $2\pi r$. Finally, we can obtain the probability distribution $P(y,\ell)$ of microtubules with minus ends at a position $y$ and length $\ell$ as:

$$P(y,\ell) = \frac{\rho(y,\ell+y)}{\int_0^\infty \rho(y,\ell+y)d\ell}. \tag{6}$$

In the particular case of monopolar spindles, since microtubule transport is inhibited $P(y,\ell) \equiv P(\ell)$ and the microtubule length distribution does not depend on the position of the minus ends. This was experimentally verified and it is shown in **Figure 2—figure supplement 1**.

## Appendix 2

DOI: https://doi.org/10.7554/eLife.31149.029

# Physical model for autocatalytic microtubule nucleation

We consider a simplified model of autocatalytic microtubule nucleation. We define the microtubule bound and unbound populations of active nucleators (or active SAFs) as $n_b(x,t)$ and $n_u(x,t)$, respectively. When unbound, active nucleators can diffuse with diffusion coefficient $D$ and become inactive with rate $k_0$. Active nucleators can bind to microtubules with rate $k_b$ and unbind with rate $k_u$. A bound nucleator, can nucleate a microtubule from a pre-existing microtubule with rate $k_{\mathrm{bra}}$. The average position at which an active nucleator will bind to a microtubule can be estimated as follows: We assume that nucleators can bind anywhere along a microtubule. Since our measurements showed that microtubule lengths are exponentially distributed with an average length $\ell$ (**Figure 2B**), a nucleator will bind on average at a distance $\ell$ from the minus end of the mother microtubule. Thus, the average location of nucleation coincides with the average location of the plus end of the mother microtubule. The density of microtubule plus ends is transported at the polymerization velocity $v_p$ (**Dogterom and Leibler, 1993**). Finally, we define the microtubule mass density as $\rho(x,t) = \int_0^\infty \tilde{\rho}(x,l,t)dl$, where $\tilde{\rho}$ is the distribution of microtubules with length $l$ at time $t$ and plus ends at $x$. In our simplified description microtubules grow at velocity $v_p$ and disappear after a characteristic turnover rate $\Theta$. We want to highlight the difference between $v_p$ and the front velocity of the growing structure. The latter depends on $k_{\mathrm{bra}}$ and vanishes in the absence of autocatalytic nucleation (**Figure 4—figure supplement 1**). Given the previous considerations, the dynamics of the system read:

$$\partial_t n_u = D\nabla^2 n_u - k_b\ell_b n_u\rho + k_u n_b - k_0 n_u \tag{7}$$

$$\partial_t n_b = k_b\ell_b n_u\rho - k_u n_b \tag{8}$$

$$\partial_t \rho = -\boldsymbol{v}_p \cdot \nabla\rho + k_{\mathrm{bra}} n_b - \Theta\rho \tag{9}$$

where $\ell_b$ is a characteristic binding length scale for the active nucleators. Next, we will consider a one-dimensional problem with the spatial coordinate $x$ being the radial coordinate from the center of the monopolar spindle. A more involved two-dimensional description of the problem is found to lead to similar results. Unbound nucleators are assumed to be activated with constant rate $\Gamma$ at the surface of chromatin in the center of the monopole ($x = 0$) (see **Figure 4A**). This leads to a boundary condition for the flux of active nucleators at the chromosomes which is expressed as $-D\partial_x n_u|_{x=0} = \Gamma$. At steady state, **Equation 8** leads to $n_b(x) = \ell_0 n_u(x)\rho(x)$, where $\ell_0 \equiv \ell_b k_b/k_u$. Using the last expression into **Equation 7** and the boundary condition at $x = 0$, at steady state we obtain:

$$n_u(x) = Ae^{-x/\ell_u} \tag{10}$$

where $A = \frac{\Gamma}{\sqrt{Dk_0}}$ is the amplitude of the gradient, proportional to the rate of activation $\Gamma$ at the chromosomes, and $\ell_u \equiv \sqrt{D/k_0}$ is the characteristic length scale of the gradient of unbound active nucleators. Finally, by using **Equation 9** we find the steady state microtubule density:

$$\rho(x) = \lambda(x)e^{-x/\ell} \tag{11}$$

where $\lambda(x) = \rho(0)\exp\left[\alpha(1 - e^{-x/\ell_u})\right]$ is a lifetime-independent function, $\rho(0)$ is the density of microtubules at $x = 0$ and $\alpha \equiv \frac{\Gamma\ell_0 k_{\mathrm{bra}}}{vk_0}$ is a dimensionless parameter. Since only bound nucleators can nucleate new microtubules, the nucleation process requires an initial source of microtubules acting as seeds for the autocatalytic growth. In our simplified model, this initial source corresponds to the boundary condition $\rho(0)$. The origin of these seed microtubules could be due to spontaneous microtubule nucleation in the cytoplasm, centrosomes, or RanGTP-mediated nucleation in close proximity to chromosomes. One possibility is that the concentration of RanGTP close to the site of its production at the chromosomes is high

enough to trigger spontaneous nucleation in the cytoplasm similar to the the case when constitutively active RanQ69L is added at sufficiently high concentration to the extract without pre-existing microtubule structures (**Figure 3B**). However, our measurements on microtubule nucleation in control and MCAK-inhibited structures shows that microtubule-independent microtubule nucleation is not sufficient to explain spindle growth or the spatial dependence of microtubule nucleation, and that microtubule independent nucleation can be accounted for as a boundary condition, suggesting that it is very localized in space.

There are two main length scales in the system: $\ell_u$ which is dictated by the gradient of unbound active nucleators and does not depend on microtubule lifetime, and $\ell$ which is the mean microtubule length. From our results, the inhibition of the motor protein MCAK affects the lifetime $\Theta$ and length $\ell$ (see **Figure 2B and C**), thus not changing $\lambda(x)$. Therefore, the model predicts that if the control monopole profile is given by $\rho_C(x) = \lambda(x)e^{-x/\ell_C}$, the perturbed monopole profile reads $\rho_M(x) = \lambda(x)e^{-x/\ell_M}$. The ratio of the two profiles follows:

$$\frac{\rho_C(x)}{\rho_M(x)} = \exp\left[x\left(\frac{1}{\ell_M} - \frac{1}{\ell_C}\right)\right]. \tag{12}$$

In **Figure 4B**, the MCAK-inhibited microtubule profile is fitted to **Equation 11** with fitting parameters $\alpha$ and $l_u$ (notice that $\rho(0)$ only rescales the arbitrary amplitude), while the parameter $\ell_M$ is measured from laser ablation measurements in MCAK-inhibited monopoles. Conversely, the microtubule density profile for control monopoles is predicted by using **Equation 11** or **Equation 12** without the need of any fitting parameter, taking $\ell_C$ as the measured mean microtubule length from control monopoles. Finally, the fits on **Figure 4D** for the microtubule nucleation profiles are done using the expression for $n_b(x)$ and adjusting the prefactor. A summary of the parameters used in the model and the procedure used to obtain them is specified in **Table 2**.

