## [Decision Letter]

Thank you for submitting your article "Autocatalytic microtubule nucleation determines the size and mass of spindles" for consideration by *eLife*. Your article has been reviewed by two peer reviewers, and the evaluation has been overseen by Andrea Musacchio as the Senior and Reviewing Editor. The reviewers have opted to remain anonymous.

The reviewers have discussed the reviews with one another and the Reviewing Editor has drafted this decision to help you prepare a revised submission.

Summary:

In this paper, the authors use mono polar spindles (obtained for instance by Eg5 knockdown) to evaluate the position of minus-ends in *Xenopus* egg extract spindles, and thus to evaluate the mechanism of microtubule nucleation, and its effect on spindle size. The authors show that the monopolar spindle density profile can be understood as an autocatalytic mechanism that is regulated by a gradient of active nucleators. This is achieved by laser ablation experiments triggering a wave of depolymerisation. Measurements of the properties of the depolymerisation wave profile allow to access quantitative information on the nucleation profiles and length distribution of the microtubules. Comparison between control spindles and spindles with MCAK inhibition, which have longer and longer-lived microtubules, is consistent with the regulated autocatalytic mechanism. The paper concludes with a computational model to account for the control and MCAK inhibition conditions. The model is well explained and nicely done, and makes a strong contribution to the paper. In conclusion, the reviewers consider this an interesting and elegant study. However, they also identify the following important points that should be addressed in a revised version of the manuscript.

Essential revisions:

1) A table of parameters, together with values that are determined in the study, should be added. For instance, it is unclear what parameters are obtained from the model fit in Figure 4.

2) While the theoretical modelling seems generally reasonable, some aspects remain unclear. Before Equation 7 the authors argue that since a nucleator can bind anywhere on microtubules with average length l, they bind on average at distance l from the end of the mother microtubule – why is it not l/2 then? They then argue that the center of mass velocity is given by v=l theta with theta the turn-over rate. Can the authors explain in more details where this relation comes from? In particular, one reviewer is surprised that the branching rate does not enter this formula; is it the case that there is still a velocity at 0 branching rate for instance?

3) In Figure 3, it is unclear from the legend what exactly is being plotted. The authors should clarify this point.

4) Since only bound nucleators are nucleating new microtubules, it seems that the whole process would not start without the formation of initial microtubules by a process distinct from the autocatalytic mechanism described in the paper. Can the authors discuss what may control the density at x=0? Is it different in the control and MCAK-inhibited conditions?

5) After Equation 1, the sentence "this rate has a very well peak that moves as a function of time, Figure 1" is confusing. The rate *Ã(t, r*) is already integrated with respect to distance away from the cut, so presumably Figure 1 does not directly inform about the shape of the function as a function of *r*. Is this correct? If not, please explain.

6) In Figure 2, the x-axis says "distance from the central"; do the authors actually mean "microtubule length L"?

7) After Equation 6, the authors mention that *P(y, l*) does not depend on *y*. Is that something that can be verified from experimental measurements?

8) In Figure 1, the two nucleation profiles for control and a-MACK conditions are plotted in arbitrary units. Are these arbitrary units defined in a consistent way, such that the difference in magnitude of the two plots is meaningful? If so, it would be helpful to clarify this in the manuscript.

9) Throughout the paper, the authors generalize the results to "mitotic spindles" or in some cases "large spindles". However, the mechanisms described here may be particular to *Xenopus* egg extract spindles, in which branched microtubule nucleation has been observed. The authors should be careful either to limit conclusions to the system described here, rather than to all mitotic spindles in general, or to explain how the results may apply in other mitotic spindles, which may not show branched microtubule nucleation, and/or where microtubule nucleation may occur exclusively at spindle poles or centrosomes.

10) The authors state that the location of minus ends in the monopolar spindles exactly corresponds to the location of microtubule nucleation, because microtubule transport is completely eliminated in the absence of Eg5. This is a major assumption in the paper, and experimental verification that no microtubule transport occurs in the absence of Eg5, and that the minus ends are exact readouts for the location of microtubule nucleation, is important.

11) The authors state that their method of laser cutting and analysis resolves the minus end locations with a single laser cut, allowing the authors to measure the "microtubule nucleation profile" (units a.u./length2). It would be helpful to provide a bit more detail regarding the method in the main text and so that the reader can follow the experiment, and, in addition, a better definition and description of "microtubule nucleation profile" would allow the reader to better understand the results in Figure 1. Right now, these key results are too briefly explained in the main text, and not understandable.

12) The authors measured microtubule nucleation under conditions in which MCAK was inhibited, and found that the number and spatial spread of nucleated microtubules was increased. From these experiments, it was concluded that microtubule nucleation depended upon the presence and dynamics of microtubules. However, can the authors exclude that the nucleation process itself is not altered by MCAK inhibition, such that stabilization of microtubule nuclei by inhibition of MCAK would lead to increased nucleation, similar to but not dependent on the presence of additional microtubule density in the presence of MCAK?

13) The description of the experiments in Figure 2, in which an "inert obstacle" was added to block microtubule polymerization, should be better detailed in the main text. Further, while the one sample image is convincing, there was no quantification of this result, nor detail on how the location of the obstacle relative to the center of the monopole would alter these results. This needs to be addressed.

14) From the RanQ69L experiments, in which new microtubules were nucleated at the edge of pre-existing structures (one experiment shown, no quantification), the authors conclude that the amount of active nucleators limit the size of monopolar spindles. This conclusion would be greatly strengthened by measuring the size of the monopoles as a function of the concentration of active nucleators.

[Editors' note: further revisions were requested prior to acceptance, as described below.]

Thank you for resubmitting your work entitled "Autocatalytic microtubule nucleation determines the size and mass of spindles" for further consideration at *eLife*. Your revised article has been favorably evaluated by Andrea Musacchio as the Senior and Reviewing Editor, and two reviewers.

The reviewers praised your effort in revising the manuscript, but there are two remaining issues that need to be addressed before acceptance, as outlined below. The required changes are textual and do not require re-review, but I kindly ask you to consider the reviewers' points very carefully:

1) One thing that remains unclear is the definition of the velocity *v*. The explanations given by the authors seem confusing, and the authors should clarify this in the manuscript, maybe with the help of a schematic. What is not clear is that (i) in their reply to the referees, the authors state that *v* is the velocity due to the branching process, but end by saying that in the absence of branching there is still a non-zero velocity. This sounds contradictory and needs to be clarified. (ii) It is unclear if this velocity is different or equal to the velocity of polymerisation of individual filaments. (iii) If the velocity is indeed associated to the microtubule mass flow as stated by the authors, it is not clear how this velocity can be independent from the branching rate. It sounds natural to think that a moving filament decorated with many branches will not create the same mass flow as an undecorated filament. These points should be clarified in the text, possibly with the aid of a schematic.

2) While the authors generalize the results to "mitotic spindles" or in some cases "large spindles", the mechanisms described here rely on branched microtubule nucleation, which, to the reviewer's knowledge, has only been observed in *Xenopus* egg extract spindles. Thus, the reviewer thinks that it is important to limit the conclusions and paper title to systems in which branched microtubule nucleation has actually been observed, since the model relies specifically on this assumption, rather than to all mitotic spindles in general. It appears that the authors address this concern by hypothesizing that branched microtubule nucleation could potentially also happen in mitotic spindles from other organisms, without supporting data. In the absence of new data to demonstrate that branching microtubule nucleation is a general mechanism across multiple organisms, this argument is not convincing, and the paper and title should be rewritten with more exact language to limit the conclusions specifically to spindles, such as the *Xenopus* egg extract spindle, in which branched microtubule nucleation has been observed.

---

## [Author Response]

Essential revisions:1) A table of parameters, together with values that are determined in the study, should be added. For instance, it is unclear what parameters are obtained from the model fit in Figure 4.

We thank the reviewers for their comments. We added two tables in the Materials and methods (Table 1 and Table 2) specifying the values obtained for microtubule dynamics measurements and the parameters used in the model, as well as the procedure used to obtain them. In Figure 4, two main parameters (𝛼 and ℓ_u_) are obtained by fitting the model to the microtubule density profile for the MCAK-inhibited monopole. A third parameter ⍴(0) which determines the amplitude of the profile is also fit to the MCAK-inhibited monopole data. Despite this amplitude is given in arbitrary units, the intensity profiles for the control and perturbed monopoles were obtained from the same extract reactions. Thus, their ratio is quantified and the same amplitude parameter ⍴(0) is used for the two different profiles. These three parameters (⍴(0), 𝛼 and ℓ_u_) obtained from the MCAK-inhibited monopole data are used to predict the control monopole microtubule density, as well as the nucleation profiles in Figure 4 (except for the prefactor in the nucleation profile that cannot be obtained from fitting the density profiles directly).These parameters are specified in Table 2.

2) While the theoretical modelling seems generally reasonable, some aspects remain unclear. Before Equation 7 the authors argue that since a nucleator can bind anywhere on microtubules with average length l, they bind on average at distance l from the end of the mother microtubule – why is it not l/2 then?

The key point is that microtubules are dynamic, and therefore the probability of an active nucleator to bind close to the minus end is larger than close to the plus end, since the binding site close to the minus end lives for longer than the binding site close to the plus end. Given that the length distribution of microtubules is exponentially distributed with mean length ℓ, this implies that the profile of active nucleators bound to a microtubule is also exponentially distributed with a length scale ℓ. Therefore, on average, active nucleators will bind at a distance ℓ from the microtubule minus end.

This calculation is slightly different from calculating the average position of a fluorescent speckle incorporated in a dynamic microtubule of average length ℓ. In this case the answer is ℓ/2 because speckles can only be incorporated at the plus end, while active nucleators can bind all along the microtubule. We made this point now clearer in the Materials and methods.

They then argue that the center of mass velocity is given by v=l theta with theta the turn-over rate. Can the authors explain in more details where this relation comes from? In particular, one reviewer is surprised that the branching rate does not enter this formula; is it the case that there is still a velocity at 0 branching rate for instance?

In our model, the velocity *v* is the speed at which the microtubule mass flows due to the branching process. This is calculated by considering the average center of mass distance between the newborn microtubule from its mother microtubule divided by the microtubule lifetime. As mentioned in Appendix 2, subsection “Physical model for autocatalytic microtubule nucleation”, since the microtubule length is exponentially distributed with length ℓ, the average branching distance is precisely ℓ. We want to highlight the difference between the microtubule mass flux velocity and the front velocity. The latter depends on the branching rate *k*_bra_ and should be zero in the absence of branching. In a similar model of autocatalytic microtubule growth, the front velocity was explicitly calculated and depended on the branching rate [see Ishihara et al. 2016]. In our case, the steady state front velocity is zero as the structures reach a finite size, although *v* is different from zero. If the branching rate is reduced, the number of branching events is reduced, but the microtubule flux for a given branching process is the same. In the absence of branching (*k*_bra_ = 0), using Equation 9 in Appendix 2, we find an exponential profile for the microtubule density with length scale ℓ, which corresponds to the case of a single microtubule at the origin.

We included some comments in the appendix highlighting the fact that *v* does not correspond to the front velocity and does not depend on the branching rate.

3) In Figure 3, it is unclear from the legend what exactly is being plotted. The authors should clarify this point.

We assume the reviewers refer to Figure 4 and not Figure 3. We first apologize for the brevity of the explanation and we agree with the reviewers that Figure 4 is somehow unclear. In this figure we show evidence supporting our model exemplified by testing Equation 12 in the Materials and methods section (or Figure 4 caption). The experimentally measured microtubule density profile of the anti-MCAK monopole is multiplied by the exponential factor exp[x(1/ℓ_M_-1/ℓ_C_)] to find the predicted control density profile according to the model, where ℓ_M_ is the average length of a microtubule in the anti-MCAK monopole and ℓ_C_ is the average length of microtubule in a control monopole. Since these two parameters are measured using laser ablation (see Table 2), the exponential factor is known and there are no fitting parameters. The blue curve shows the experimental control profile (same as in Figure 4) whereas the gray curve shows the predicted control profile, using Equation 12 together with the anti-MCAK profile from Figure 4. We included some remarks on the explanation both on the main text and on the caption of Figure 4 to better explain the plots.

4) Since only bound nucleators are nucleating new microtubules, it seems that the whole process would not start without the formation of initial microtubules by a process distinct from the autocatalytic mechanism described in the paper. Can the authors discuss what may control the density at x=0? Is it different in the control and MCAK-inhibited conditions?

We agree with the reviewers that autocatalytic microtubule nucleation requires initial seed microtubules to trigger the growth of the structure. This initial density corresponds to ⍴(0). As suggested by the reviewers the mechanism setting ⍴(0) is different than autocatalytic nucleation. The origin of these seed microtubules can be due to spontaneous microtubule nucleation in the cytoplasm, centrosomes, or RanGTP-mediated nucleation in close proximity to chromosomes. One possibility is that the concentration of RanGTP close to the site of its production at the chromosomes is high enough to trigger spontaneous nucleation in the cytoplasm similar to the case when constitutively active RanQ69L is added at sufficiently high concentration to the extract without pre-existing microtubule structures. However, our work shows that microtubule-independent microtubule nucleation is not sufficient to explain spindle growth or the spatial dependence of microtubule nucleation, and that microtubule independent nucleation can be accounted for as a boundary condition, suggesting that it is very localized in space. We added this discussion in the model section of Appendix 2.

Regarding the microtubule density at *x=0,* i.e. ⍴(0), it is the same for both profiles and it can be obtained by fitting the model to one of the experimentally measured microtubule density profiles (see Table 2). In particular, in our analysis we fit this value in the MCAK-inhibited profile and use it to predict the measured control profile. Therefore, our model is consistent with the microtubule density profile being the same for the control and perturbed case at the origin *x=0*. We have better emphasized this point in the main text.

5) After Equation 1, the sentence "this rate has a very well peak that moves as a function of time, Figure 1" is confusing. The rate Ã(t, r) is already integrated with respect to distance away from the cut, so presumably Figure 1 does not directly inform about the shape of the function as a function of r. Is this correct? If not, please explain.

The reviewers are correct, there was a confusion on the definition of *A(t,r)*. This quantity is integrated over the angular and radial coordinates. We corrected this part in Appendix 1 subsection “Laser ablation method”.

6) In Figure 2, the x-axis says "distance from the central"; do the authors actually mean "microtubule length L"?

Thank you for pointing this out. This is a typo that we fixed in the new version.

7) After Equation 6, the authors mention that P(y, l) does not depend on y. Is that something that can be verified from experimental measurements?

As mentioned in Equation 6 in Appendix 1, given that microtubule transport is inhibited, the microtubule length distribution in monopoles should theoretically not depend on the position of the minus-ends *y*. This is in contrast to the microtubule length distribution in spindles, where there exists a clear dependence of the mean microtubule length along the spindle long axis due to microtubule transport, and becomes homogenous upon inhibition of microtubule transport (see Brugués et al. 2012). We verified this experimentally and we now included a new supplementary figure (Figure 2—figure supplement 1) showing how *P(y,*ℓ) is independent of *y*.

8) In Figure 1, the two nucleation profiles for control and a-MACK conditions are plotted in arbitrary units. Are these arbitrary units defined in a consistent way, such that the difference in magnitude of the two plots is meaningful? If so, it would be helpful to clarify this in the manuscript.

Yes, these arbitrary units are consistently defined by using angular fluorescence (average microtubule density) profiles of control and MCAK-inhibited structures from the same extract reaction to determine the ratio between these two and enable a reliable comparison. We added this clarification in the caption of the corresponding figure (Figure 2).

9) Throughout the paper, the authors generalize the results to "mitotic spindles" or in some cases "large spindles". However, the mechanisms described here may be particular to Xenopus egg extract spindles, in which branched microtubule nucleation has been observed. The authors should be careful either to limit conclusions to the system described here, rather than to all mitotic spindles in general, or to explain how the results may apply in other mitotic spindles, which may not show branched microtubule nucleation, and/or where microtubule nucleation may occur exclusively at spindle poles or centrosomes.

We agree with the reviewers that such mechanism may not apply to all types of mitotic spindles, especially spindles with a small amount of microtubules (~10) as in the case of fission yeast. However, we argue that several evidences support the fact that this mechanism might apply to the majority of eukaryotic mechanisms, where the number of microtubules forming the spindles is on the order of 10^4^-10^5^ microtubules, and in spindles that are larger than the length of individual microtubules. Components involved in microtubule branching (e.g., augmin) and in the Ran pathway (e.g., RCC1) have been identified in many eukaryotic organisms (mainly metazoans) [Dasso, 2002; Hsia et al. 2014; Sánchez-Huertas and Lüders, 2015; Kamasaki et al. 2013; Savoian and Glover 2014; Goshima et al.2008]. Moreover, microtubule-dependent microtubule nucleation has been recently proposed to account for size regulation in human tissue culture cells [Oh et al. 2016]. Therefore, we hypothesize that spatially regulated autocatalytic microtubule nucleation could also apply to other organisms, especially those with spindles consisting of more than a few thousand microtubules and lengths larger than their microtubule average length. To validate this hypothesis, further experiments in other organisms will need to be performed. We now included some elements of the previous discussion at the end of the manuscript.

10) The authors state that the location of minus ends in the monopolar spindles exactly corresponds to the location of microtubule nucleation, because microtubule transport is completely eliminated in the absence of Eg5. This is a major assumption in the paper, and experimental verification that no microtubule transport occurs in the absence of Eg5, and that the minus ends are exact readouts for the location of microtubule nucleation, is important.

We performed a quantitative analysis of the velocity distribution of the Video 2 showing that tubulin speckles do not significantly move in monopoles. We added this analysis as a new panel in Figure 1—figure supplement 1. Tracking individual tubulin speckles we evaluated the velocity distribution along the *x* and *y*-components given a certain cartesian coordinate frame. We obtained 0.09 ± 0.03 μm/min in the *x*-component and 0.08 ± 0.04 μm/min in the *y*-component (mean ± SDM, n = 717 speckles). The mean velocity of the speckles was found to be 0.13 ± 0.08 μm/min, which is negligible compared with the microtubule transport in spindles (Sawin et al., 1992 (2.10 ± 0.36 µm/min); Maddox et al., 2003 (2.3 ± 0.6 µm/min); Miyamoto et al., 2004 (around 2 µm/min); Cameron et al., 2011 (around 2.5 µm/min in spindle center); Brugués et al., 2012 (2.5 µm/min)). Furthermore, to our knowledge, the maximum reported minus-end polymerization velocity in *X. laevis* extract is 0.3 μm/min [Hendershott *et al.* 2014]. This speed is negligible compared to the polymerization dynamics of the plus ends (v_p_≈ 24 µm/min, v_d_≈ 37 µm/min), and would imply an error of 100 nm over the average lifetime of a microtubule (~20 s). Additionally, the videos of the cuts are of the order of 30 s and therefore, the uncertainty in the cut position after the data analysis is already larger than the error minus-end polymerization/depolymerization would introduce.

11) The authors state that their method of laser cutting and analysis resolves the minus end locations with a single laser cut, allowing the authors to measure the "microtubule nucleation profile" (units a.u./length2). It would be helpful to provide a bit more detail regarding the method in the main text and so that the reader can follow the experiment, and, in addition, a better definition and description of "microtubule nucleation profile" would allow the reader to better understand the results in Figure 1. Right now, these key results are too briefly explained in the main text, and not understandable.

We agree with the reviewers that the definition of the microtubule nucleation profile was not clearly stated in the main text and we included now the explicit definition. We define the microtubule nucleation profile at a distance *r* from the center of the monopole as the number of minus ends per unit length at *r* divided by 2π*r*. We still prefer to keep the laser ablation method in the supplement since we believe that properly explaining it would require a long discussion that would interfere with the readability of the manuscript. However, we have added a few more sentences in the main text so that the reader can understand the essential elements of the method.

12) The authors measured microtubule nucleation under conditions in which MCAK was inhibited, and found that the number and spatial spread of nucleated microtubules was increased. From these experiments, it was concluded that microtubule nucleation depended upon the presence and dynamics of microtubules. However, can the authors exclude that the nucleation process itself is not altered by MCAK inhibition, such that stabilization of microtubule nuclei by inhibition of MCAK would lead to increased nucleation, similar to but not dependent on the presence of additional microtubule density in the presence of MCAK?

One possibility is that MCAK-inhibition could by itself increase nucleation independently of microtubules. However, this would only lead to an overall increase of microtubule nucleation, which alone would not be sufficient to account for the dramatic change in the spatial dependence of the nucleation profile we observe in Figure 2. Thus, microtubule nucleation in these structures depends on the presence and dynamics of microtubules. An additional evidence supporting the fact that MCAK inhibition is not affecting the overall nucleation amplitude is that the microtubule density at the origin ⍴(0) is found to be the same for the control and perturbed monopoles. We added this discussion in the main text.

13) The description of the experiments in Figure 2, in which an "inert obstacle" was added to block microtubule polymerization, should be better detailed in the main text. Further, while the one sample image is convincing, there was no quantification of this result, nor detail on how the location of the obstacle relative to the center of the monopole would alter these results. This needs to be addressed.

We added an additional panel in Figure 3 in the main text together with a new video. The new panel in the main text shows the quantification of how an obstacle affects the MT density profiles, at distances ranging from 11 to 30 µm from the center. The analysis considers n=6 obstacles and shows a clear valley at the obstacle position which corresponds to the casted shadow observed in the videos. We plan to further investigate how the obstacle position alters the results in a follow up paper; however, our present data do not show a dependence on the distance. We added a video that shows a monopole growing against an obstacle (Video 7).

14) From the RanQ69L experiments, in which new microtubules were nucleated at the edge of pre-existing structures (one experiment shown, no quantification), the authors conclude that the amount of active nucleators limit the size of monopolar spindles. This conclusion would be greatly strengthened by measuring the size of the monopoles as a function of the concentration of active nucleators.

The main reason to perform these experiments was to show that the activation of more nucleators leads to outgrowth of pre-existing structures, which suggests that the amount of active nucleators is limiting in normal spindles. As suggested by the reviewers, we have added a new panel in Figure 3 from the main text that quantifies the growth of the monopole displayed in the same figure.

Unfortunately it is challenging to measure the size of monopoles as a function of RanQ69L concentration because it is difficult to measure a steady state size. This is a consequence of the appearance of mini asters at some point during the monopole growth that presumably sequester nucleators or other components (such as tubulin or MAPs). When we did a titration of RanQ69L using the same extract reaction we made the following observations:

60 µM: Spontaneous nucleation occurred everywhere in the cytoplasm and motor action immediately formed mini Ran asters (similar to the control image and movie shown in Figure 3 left and Video 8). The density of Ran asters was high enough that they dragged out microtubules from the pre-existing monopole such that they did not grow but instead got ripped apart within ~15 min.

30 µM and 15 µM: Monopoles grew until Ran asters appeared everywhere. The higher the RanQ69L concentration the earlier Ran asters appeared. Then the growth of monopoles stopped, presumably due to either depletion of nucleators in the mini asters, or some other component (e.g., tubulin or MAPs).

10 µM: This is the concentration shown in Figure 3 right. For this preparation of extract Ran asters appeared very late in the cytoplasm and thus the monopole had maximal growth. We now included a quantification for this case showing how the front advances.

5 µM: For this concentration the monopoles almost did not grow. Most likely because the RanQ69L concentration was too low to effectively increase the nucleation from pre-existing monopoles. No Ran asters were formed in the cytoplasm.

Finally, it is worth mentioning that other authors also reported difficulties and variability between extracts in RanQ69L experiments [Maresca et al. 2009]. Despite the difficulties in quantifying the titration of the RanQ69L experiments, the conclusions of this experiment remain the same, i.e., increasing the amount of active nucleators leads to the growth of the monopolar structure, implying that active nucleators limit the growth of these structures.

[Editors' note: further revisions were requested prior to acceptance, as described below.]

The reviewers praised your effort in revising the manuscript, but there are two remaining issues that need to be addressed before acceptance, as outlined below. The required changes are textual and do not require re-review, but I kindly ask you to consider the reviewers' points very carefully:1) One thing that remains unclear is the definition of the velocity v. The explanations given by the authors seem confusing, and the authors should clarify this in the manuscript, maybe with the help of a schematic. What is not clear is that (i) in their reply to the referees, the authors state that v is the velocity due to the branching process, but end by saying that in the absence of branching there is still a non-zero velocity. This sounds contradictory and needs to be clarified. (ii) It is unclear if this velocity is different or equal to the velocity of polymerisation of individual filaments. (iii) If the velocity is indeed associated to the microtubule mass flow as stated by the authors, it is not clear how this velocity can be independent from the branching rate. It sounds natural to think that a moving filament decorated with many branches will not create the same mass flow as an undecorated filament. These points should be clarified in the text, possibly with the aid of a schematic.

We have clarified the confusion between polymerization velocity and front velocity in the text and with a schematic (Figure 4—figure supplement 1). In our simplified description, the velocity *v* appearing in the density equation is the polymerization velocity. The front velocity is a consequence of both the polymerization of microtubules and the branching process. This distinction becomes clearer in the new figure.

2) While the authors generalize the results to "mitotic spindles" or in some cases "large spindles", the mechanisms described here rely on branched microtubule nucleation, which, to the reviewer's knowledge, has only been observed in Xenopus egg extract spindles. Thus, the reviewer thinks that it is important to limit the conclusions and paper title to systems in which branched microtubule nucleation has actually been observed, since the model relies specifically on this assumption, rather than to all mitotic spindles in general. It appears that the authors address this concern by hypothesizing that branched microtubule nucleation could potentially also happen in mitotic spindles from other organisms, without supporting data. In the absence of new data to demonstrate that branching microtubule nucleation is a general mechanism across multiple organisms, this argument is not convincing, and the paper and title should be rewritten with more exact language to limit the conclusions specifically to spindles, such as the Xenopus egg extract spindle, in which branched microtubule nucleation has been observed.

As suggested by the reviewer, we changed the title of the paper to include *Xenopus laevis* egg extract and restrict our results to this system. In the Discussion we made more explicit that the possibility that this mechanism may be shared by other organisms is hypothetical and needs to be tested.